# Do Rutin and Quercetin Retain Their Structure and Radical Scavenging Activity after Exposure to Radiation?

**DOI:** 10.3390/molecules28062713

**Published:** 2023-03-17

**Authors:** Natalia Rosiak, Judyta Cielecka-Piontek, Robert Skibiński, Kornelia Lewandowska, Waldemar Bednarski, Przemysław Zalewski

**Affiliations:** 1Department of Pharmacognosy, Poznan University of Medical Sciences, Rokietnicka 3, 60-806 Poznan, Poland; 2Department of Medicinal Chemistry, Medical University of Lublin, Jaczewskiego 4, 20-090 Lublin, Poland; 3Institute of Molecular Physics, Polish Academy of Sciences, Smoluchowskiego 17, 60-179 Poznań, Poland

**Keywords:** quercetin, rutin, electron beam irradiation, antioxidant, HPLC, EPR, FTIR

## Abstract

The influence of ionizing radiation on the physicochemical properties of quercetin and rutin in the solid state was studied. Quercetin and rutin were irradiated with the standard recommended radiation dose (25 kGy) according to EN 522 standard. The samples were irradiated by electron beam radiation. EPR studies indicate the formation of a small number of free radicals due to irradiation. Moreover, some radicals recombined with the mean lifetime of 1200 and 93 h, and a stable radical concentration reached only 0.29 and 0.90 ppm for quercetin and rutin, respectively. The performed spectroscopic study (FT-IR) confirmed the radiostability of the flavonoids tested. Chromatographic tests (HPLC, HPLC-MS) showed that irradiation of quercetin and rutin with a 25 kGy dose did not change the physicochemical properties of the tested compounds. Degradation products were not observed. The antioxidant activities were determined by the 2,2-diphenyl-1-pycrylhydrazyl (DPPH) free radical scavenging activity assay, ABTS Radical Scavenging Assay (ABTS), Ferric Reducing Antioxidant Power Assay (FRAP), Cupric Ion Reducing Antioxidant Capacity Assay (CUPRAC). The conducted research confirmed that exposure to ionizing radiation does not change the chemical structure of tested flavonoids and their antioxidant properties.

## 1. Introduction

Polyphenols are a class of antioxidants that are frequently recommended as promising candidates for antioxidant therapy due to their potential health properties. Literature confirmed the preventive action of polyphenols for the development of many civilization diseases, such as the decline in the incidence of cardiovascular diseases [1,2], metabolic diseases [3], or neoplastic diseases [4,5]. One of the most important properties of polyphenols is the ability to neutralize reactive oxygen species, chelate metals, and reduce metals [6,7,8].

One of the most abundant flavonoids with proven health-promoting properties are quercetin and its glucoside, rutin. Elderberries, cranberries, yellow and red onions, buckwheat, and black currants are especially rich in quercetin [9]. The sources of rutin include common rue, white mulberry, asparagus, apricots, grapes, grapefruits, oranges, and buckwheat, with buckwheat (*Fagopyrum esculentum*) being considered one of the best sources of rutin [10]. Regular consumption of rutin is connected with a protective effect on the kidneys, a therapeutic effect in the treatment of various severe vascular inflammatory diseases, and a neuroprotective effect [11,12,13].

The antioxidant activity of quercetin plays a significant role in the prevention and treatment of osteoporosis, lung, cardiovascular, and age-related diseases [14,15,16]. As a result, quercetin increases the body’s antioxidant capacity. Some studies also show that quercetin lowers the health risks connected to obesity, including high blood pressure and diabetes disease [17,18,19,20].

All this confirms that quercetin and rutin have the potential for medical application. It is important to remember that the substances used in the production of drugs and dietary supplements must not be microbiologically contaminated. Sterilization methods are used to inactivate microorganisms. An official sterilization technique of drugs that is accepted by the IAEA and listed in pharmacopeias is radiation sterilization [21]. Sterilization is necessary for pharmaceutical items in order to assure their safety and efficacy. The effect of ionizing radiation on drugs has been extensively studied for more than 20 years [21]. A range of medications, including vitamins, antiprotozoals, antibiotics, steroids, and anticancers, have been researched in relation to pharmacological sterilization [22,23,24,25,26,27]. The existing literature also provides information on the effects of ionizing radiation on plant-derived compounds and extracts [28,29,30,31,32]. Different molecular properties of drugs and natural compounds affect their radiosensitivity. Irradiation can result in a small number of radiolytic products, which may be toxic [27]. Because of this, it is important to evaluate the effects of ionizing radiation on drugs. The impact of radiation can be monitored by a variety of analytical methods, such as electron paramagnetic resonance (EPR), Fourier-transform infrared spectroscopy (FT-IR), high-pressure liquid chromatography (HPLC), and liquid chromatography-mass spectrometer (LC-MS) [23,24,33,34,35,36,37].

The effects of ionizing radiation on drugs have been studied for many years, but the topic of radiation stability of plant-derived compounds with health-promoting potential is still unfathomable. The main objective of this study was to provide evidence that radiation sterilization (electron beam radiation of 25 kGy dose) is a safe method of sterilizing rutin and quercetin so they can be used as ingredients in drugs or dietary supplements. This will provide the patient with safe and effective therapy with preparations with recognized antioxidant potential.

## 2. Results

### 2.1. EPR

EPR spectra of non-irradiated (0 kGy) and irradiated (dose 25 kGy) samples recorded at 51 and 595 h (quercetin), 50 and 596 h (rutin) after irradiation are shown in Figure 1a,b, respectively.

With the passage of time after irradiation, there is a clear decrease in the intensity of the spectrum for rutin (Figure 1b) and only a slight decrease for quercetin (Figure 1a). The above fact indicates that at least some of the free radicals formed as a result of radiation are unstable. Figure 1c,d presents the concentration of free radicals vs. time after irradiation (25 kGy) calculated from EPR spectra for quercetin and rutin, respectively.

The total concentration of free radicals *C_tot_(t)* at any time *t* after irradiation was described by Equation (1) [28]:(1)$$C_{tot}(t)=C_{s}+C_{u}e^{-\frac{t}{T}}$$
where *C_s_*—the concentration of stable radicals, *C_u_*—the concentration of unstable free radicals, and *T*—the mean lifetime of unstable radicals. Approximation of Equation (2) to the experimental points gives the following parameters for quercetin/rutin, respectively: *C_tot_*(*t* = 0) = 0.42 ± 0.13/2.42 ± 0.46 ppm, *C_s_
*= 0.29 ± 0.06/0.90 ± 0.09 ppm, *C_u_
*= 0.13 ± 0.07/1.52 ± 0.3 ppm. The radical mean lifetime *T* of unstable radicals for irradiated quercetin is 1200 ± 900 h and 93 ± 32 h for rutin. EPR spectra of non-irradiated samples consist only of signals from stable radicals, and the concentration is *C_s_* = 0.22 ± 0.07/0.28 ± 0.06 ppm for quercetin/rutin, respectively. Note that spectra presented in Figure 1a,b were recorded for the samples with different masses, so the spectrum amplitude does not directly correspond to the concentration of radicals in the sample.

### 2.2. Fourier Transform Infrared Spectroscopy (FT-IR) Analysis

Appendix A (see Appendix A) shows theoretical and experimental absorption spectra in IR range of quercetin. The experimental IR absorption spectra of quercetin is different from the calculation DFT spectrum. The calculated spectrum has more bands than the experimental IR spectrum but narrower bands. Because of this, one broad band observed in the experimental spectrum has more components and is related to larger amounts and type of vibration. The comparison of the frequencies calculated by the DFT–B3LYP method with the experimental values reveals the overestimation of the calculated vibrational modes due to neglect of anharmonicity in the real system.

The IR absorption spectrum of the quercetin is dominated by the strong bands at 1166, 1206, 1244, 1317, 1517, 1616, and 1656 cm^−1^. The first and third bands are mainly related to the rocking vibration of the C–H bonds, but they have an additional component corresponding to the rocking vibration of the O–H and stretching vibration of the C–O bonds, respectively. The second band is associated with the stretching vibration of the C–O bond in C–O–H group in trihydroxychromenone, the rocking vibration of the O–H and C–H bonds, and the stretching vibration of the C–O. The bands observed at 1317 cm^−1^ and 1517 cm^−1^ related mainly to the stretching vibration of the C–C and C=C bonds. However, the bands at 1616 and 1656 cm^−1^ correspond to the stretching vibration of the C=O bond. Of course, most of the bands have additional components related to the bending vibration of the C–O–H bonds and the rocking vibration of the C–H bonds (see Appendix A). The range 1100–1700 cm^−1^ also includes less intense bands, such as 1288 and 1434 cm^−1^, which correspond to the stretching vibration of the C–O bonds. The bands below 1000 cm^−1^ are observed in association with the deformation vibration of the C–H, O–H bonds and deformation and torsional of the molecules, while those above 3000 cm^−1^ are related to the stretching vibration of the C–H and O–H bonds.

In the absorption spectrum for the quercetin after irradiation, all bands recorded before the lighting are observed (see Figure 2).

However, the shapes and intensities of some bands are changing. The differences are visible to the bands associated with the stretching vibration of the C–O (1434 cm^−1^), C–C (1460 cm^−1^) bonds and rocking and deformation vibration out of the plane of the C–H (1244, 1005, 820 and 790 cm^−1^), and O–H (880 cm^−1^) bonds. Based on the chromatographic analysis, the authors excluded quercetin degradation. Observed changes may suggest that irradiation creates the non–radical form of quercetin (the creation of which is not excluded by EPR studies), which causes hydrogen bonds to change. The analysis of the FT–IR spectra of rutin was performed on the basis of the work published by Paczkowska et al. [38]. The IR absorption spectrum of the rutin is dominated by the strong bands in the range 1000–1700 cm^−1^. Stretching vibrations of the C–O bond are at 1013 and 1506 cm^−1^. In addition, in this range, there are bands with 1066 cm^−1^ (C–O stretching in heterocyclic and mannopyranosyl rings), 1088 cm^−1^ (C–O stretching in and at mannopyranosyl ring), 1205 cm^−1^ (C–C stretching in the heterocyclic ring), 1296 cm^−1^ (C–C–H bending in all rings + C–H rocking at all rings), 1363 cm^−1^ (C–H rocking at mannopyranosyl ring), 1453 cm^−1^ (C–C–H bending at dihydroxyphenyl ring + C–O–H bending at benzene ring), 1573 cm^−1^ (C=C stretching in benzene, heterocyclic and dihydroxyphenyl rings), 1607 cm^−1^ (C=C s in benzene and heterocyclic rings), and 1656 cm^−1^ (C=O stretching at the heterocyclic ring) [38]. The intense band observed at 3404 cm^−1^ corresponds to the stretching vibration of the O–H bonds.

In the absorption spectrum for the rutin after irradiation, all bands recorded before the lighting are observed (see Figure 3). The lack of visible changes in the spectrum of rutin after irradiation confirms the lack of influence of electron beam radiation (dose 25 kGy) on the structure of the compound.

### 2.3. High-Performance Liquid Chromatography (HPLC) and Liquid Chromatography–Mass Spectrometry (LC-MS) Analysis

The studies showed that rutin and quercetin were resistant to radiation at a dose of 25 kGy. No degradation product was observed by HPLC and LC-MS after 25 kGy electron beam irradiation (see Appendix A). Analysis of the HPLC-MS results showed that rutin is slightly contaminated with one impurity with molecular ion formula C_21_H_19_O_12_ (see Appendix A). This impurity is hyperin, which is an active compound that occurs in plant raw materials along with rutin.

### 2.4. Antioxidant Properties before and after Sterilization

After exposition under electron beam radiation, quercetin and rutin were studied in regard to the influence of irradiation on their antioxidant properties. The bar graphs (Figure 4) show the IC_50_ values for the DPPH and ABTS assay and the IC_0.5_ values for the CUPRAC and FRAP assay.

Results showed a decrease in antioxidant properties in the CUPRAC test and no changes in the DPPH, ABTS and FRAP tests for 25 kGy quercetin. The changes seen in the antioxidant activity of rutin and 25 kGy rutin oscillate within the limits of error. This indicates that a dose of 25 kGy had no effect on the properties of this compound.

## 3. Discussion

Many studies refer to the effect of radiation sterilization on drugs [39,40,41], especially antibiotics [22,23,24,25,26]. The need to search for natural alternatives that exhibit health properties has increased interest in polyphenols and plant raw materials containing active compounds with antioxidant properties. For this reason, the use of plant materials and polyphenols as food ingredients or dietary supplements has increased in recent years [42,43,44,45].

The aim of our research was to show that quercetin and rutin can be markers indicating the radiostability of plant raw materials. Both of the tested compounds are characterized by biological activity related to the neutralization of free radicals. Therefore, in our research, we chose the assessment of antioxidant properties as a study of the effects of possible changes after irradiation. The difference in the chemical structure of quercetin and rutin also makes it possible to assess the influence of the presence of sugar on the radiopersistence of the tested flavonoids. Unfortunately, the EPR spectra of rutin and quercetin consist of single lines with no clearly visible hyperfine or super-hyperfine structure. Therefore, from EPR studies, it is impossible to determine the place in the molecule where the bond breaks due to irradiation. Since only the intensity of the line changes versus time (the line shape is unchanged) remains after irradiation, it can be assumed that stable and unstable radicals have an identical structure. The instability of some radicals, especially those close to the surface, may result from their interaction with reactive oxygen species in the air. EPR measurements have shown that the studied samples are very resistant to radiation, and that the dose of 25 kGy causes the formation of radicals with a total concentration of 0.42 and 2.42 ppm for quercetin and rutin, respectively. Comparing the concentrations of stable free radicals in irradiated/non-irradiated samples, it can be concluded that irradiation does not produce additional stable radicals in the case of quercetin and creates much less than 1 ppm for rutin, within the limits of experimental error.

Analysis of the FT-IR spectra of rutin exposed to the 25 kGy dose of electron beam confirmed no change in the nature of the bands (see Figure 3). In the case of FT-IR spectra for quercetin after irradiation, the registered changes can be associated with the formation of free radicals under the influence of an electron beam (see Figure 2). It is worth paying attention to the work of Jiang et al. [46], which describes the effect of ionizing radiation on the production of reactive oxygen species by flavonoids. Their published results confirm that both quercetin and its aglycone rutin are capable of producing reactive oxygen species under the influence of ionizing radiation.

The researchers emphasize that the presence in the quercetin structure of the 3–OH group, four additional aromatic hydroxyl groups, and a double bond reduces the energy required for the formation of $O_{2}^{•-}$ under the influence of ionizing radiation.

The lack of guidance changes in the structure of quercetin and rutin is also confirmed by the results of HPLC and HPLC-MS, which exclude the possibility of the formation of degradation products. We do not observe any shifts in the retention times before and after irradiation. The obtained results confirm the radiodurability of the tested flavonoids.

In all the antioxidant tests performed, 0 kGy quercetin has higher properties than 0 kGy rutin. It is directly related to the structure of the studied flavonoids. The rutin is made up of two fragments, i.e., aglycone, which is quercetin linked by a glycosidic bond with the disaccharide rutinose.

Rutinose is composed of 1-rhamnopyranose (6-deoxy-1-mannose) combined with d-glucopyranose through an α- (1 → 6) -O-glycosidic bond. The presence of this disaccharide changes the properties of the compound in relation to quercetin [47]. As reported in the literature, the antioxidant activity in the DPPH and ABTS tests is significantly influenced by the 4-OH group and the catechol group in the B ring and the presence of hydroxyl groups in the A ring [48]. In the case of the FRAP test, the activity of iron reduction is influenced by the ortho-dihydroxy substitution in the B-ring and the 3–OH group in the C-ring [49]. This is very clearly visible in the case of quercetin and rutin. The 3–OH moiety present in the quercetin structure is responsible for keeping the B ring in the same plane as the A and C rings. This is due to the interaction of 3–OH with the B ring through a hydrogen bond with 6′–H. In the case of rutin at position 3, quercetin is linked by a glycosidic bond with the disaccharide rutinose. Previous studies have confirmed a decrease in rutin activity, which is probably caused by the loss of the B–ring coplanar with the rest of the molecule [50].

The study of the antioxidant properties of flavonoids after exposure to the electron beam indicates a decrease in antioxidant properties in the CUPRAC test and no changes in the DPPH, ABTS and FRAP tests for 25 kGy quercetin. The changes in antioxidant activity of irradiated quercetin may be attributed to the free radicals generated by electron beam radiation. For example, the influence of free radicals on the antioxidant properties of polyphenols was observed by Rosiak et al. in the case of resveratrol [28]. In the case of the 25 kGy rutin, we observe changes that oscillate around the measurement error. Therefore, it can be indicated that a dose of 25 kGy had no effect on the properties of this compound. In the literature, there are reports on the lack of changes or the presence of changes in antioxidant properties for structures containing aromatic rings. For instance, Lampart-Szapa et al. observed that the majority of lupin extracts’ antioxidant efficacy was reduced when irradiation dosages were raised [31]. This effect was also noted by Al-Kuraieef et al. in the case of the methanolic extract of thyme [32]. Other research does not reveal any impact of radiation doses 5 and 25 kGy on the antioxidant activity of cinnamon compounds [30].

## 4. Materials and Methods

### 4.1. Materials

With Sigma Aldrich (St. Louis, MO, USA) were supplied: rutin, quercetin, potassium bromide (KBr), 2,2-Diphenyl-1-picrylhydrazyl (DPPH), FeCl_3_∙6H_2_O, 2,4,6-Tri(2-pyridyl)-s-triazine (TPTZ)ascorbic acid, neocuproine. Ammonium acetate (NH_4_Ac) and methanol was supplied by Chempur (Piekary Śląskie, Poland). With POCH (Gliwice, Poland) were supplied: CuCl_2_∙2H_2_O, acetic acid (99.5%), ethanol (96%), sodium acetate trihydrate (C_2_H_3_NaO_2_∙3H_2_O), glacial acetic acid. Acetonitrile of an HPLC grade was supplied by Romil (Waterbeach, Cambridgeshire, England). High-quality pure water was prepared using a Direct-Q 3 UV purification system (Millipore, Molsheim, France, model Exil SA 67120). 

### 4.2. Methods

#### 4.2.1. Irradiation

Around 0.5 g of rutin and quercetin was added to a 6 mL of colorless sodium glass vials, which was then sealed with a plastic stopper. The vials had a wall thickness of 1 mm, a 15 mm diameter, and a 55 mm height. The vials were irradiated to a 25 kGy dose of electron beam radiation (NIIEFA, St. Petersburg, Russia) on behalf of the Radiation Sterilization Plant of Medical Devices and Allografts. Parameters: set dose is 25 kGy, the transporter is 0.621 m∙min^−1^, set current is 500 mA, energy is 10 MeV, the calibration factor is 15.5 and the sampling time is 0.3 s.

#### 4.2.2. Electron Paramagnetic Resonance (EPR) Spectroscopy

X- band (≈9.4 GHz) EPR spectra were recorded at room temperature using a Bruker ELEXSYS 500 spectrometer (Bruker, Billerica, MA, USA). Irradiated rutin and quercetin powder samples were inserted in quartz tubes (Wilmad) that were placed into the resonator. We used low microwave power to record the spectra to avoid saturation of the EPR lines and each spectrum was accumulated 10 times due to the low EPR signal from the radicals. The concentration of free radicals was obtained by comparing the spectra of rutin and quercetin with the spectrum of a standard (Al_2_O_3_:Cr^3+^) with a known number of unpaired electrons. A more detailed description of the procedure for determining the concentration of radicals is elsewhere [28,51].

#### 4.2.3. Fourier Transform Infrared Spectroscopy (FT-IR)

Potential changes in the stability of irradiated and non-irradiated samples of rutin and quercetin were examined by spectroscopy using a Fourier transform infrared (FT-IR) spectrometer (Bruker Equinox 55 spectrometer, Bruker Optics, Ettlingen, Germany). Samples were prepared with potassium bromide as a matrix material and were mixed in proportions of 1 mg of rutin/quercetin sample (0 kGy/25 kGy) to 200 mg KBr. Pellets were formed under 10 ton∙cm^−2^ of pressure with a barrel of 13 mm in diameter. Absorption spectra were measured with a resolution of 4 cm^−1^ within a wavenumber range from 4000 to 400 cm^−1^ (400 scans per spectrum) and with pure KBr pellet as a blank sample. All measurements were carried out at room temperature.

#### 4.2.4. Computation

The B3LYP hybrid functional and 6-311G(d,p) basis set were used in the density functional theory (DFT) technique to improve the molecular geometries. Additionally, calculations of the normal mode frequencies and intensities were made. A Gaussian 09 package (Wallingford, CT, USA) was used for all calculations [52]. The initial shape of the molecules under investigation and a visual examination of the normal modes were proposed using the GaussView (Wallingford, CT, USA, Version E01) [53] program. The vibrational calculations used a scaling factor of 0.967.

#### 4.2.5. HPLC and HPLC-MS Analysis

In the study, a HPLC Shimadzu Prominence-i LC-2030C equipped with DAD detector was used. The software was LabSolution DB/CS (version 6.50). Solutions of irradiated and non-irradiated rutin and quercetin were prepared in methanol at a concentration of 0.4 mg∙mL^−1^. The solutions thus obtained were filtered using syringe filters with a 0.45 μm filter into vials at volume 1.5 mL. The samples were chromatographed on a Kinetex, C18, 100A, 100 mm × 2.1 mm column (Phenomenex, Torrance, CA, USA) at 5 µm particle sizes. The mobile phase was acetonitrile and 0.1% formic acid (20:80 *v*/*v*) filtered through a 0.22 μm nylon membrane and ultrasonically degassed prior to use. The flow rate of the mobile phase was 0.4 mL∙min^−1^. The injection volume was 10 µL. Chromatograms were monitored at the wavelength λ_max_ = 353 nm using the UV detector. Separation was performed at 25 ℃ and the analysis time was 15 min per sample.

An Agilent liquid chromatograph mass spectrometer (Agilent Accurate-Mass Q-TOF LC/MS G6520B system model) with dual electrospray ion source and an ultra-high pressure liquid chromatography Infinity 1290 system consisting of a binary pump (G4220A), autosampler (G4226A), thermostat (G1330B FC/ALS), DAD (G4212A), and TCC (G1316C) modules (Agilent Technologies, Santa Clara, CA, USA) were used. The MassHunter workstation software B.084.00 was used to control the system and to conduct qualitative analysis. The stationary phase was Hibar RP-18e (2.1 mm × 50 mm, 2 µm particle size) (Merck, Darmstadt, Germany). The isocratic elution with acetonitrile: 0.1% HCOOH (10:90 *v*/*v*) during 0.5 min was used; next, the gradient elution was begun within 9 min from the composition ratio (60:40). The flow rate was 0.3 mL∙min^−1^, and thermostating at 35 °C was applied. The main parameters were set as follows: MS: ESI—negative polarity, gas temperature 325 °C, drying gas 10 L∙min^−1^, nebulizer pressure 40 psig, capillary voltage 3500 V, fragmentor voltage 175 V, skimmer voltage 65 V, and octopol RF 750 V. To collect spectral data, MS / MS mode was used with the range limited to mass 90–1050 m/z and the acquisition rate: 2 spectra∙s^−1^.

#### 4.2.6. Antioxidant Assay

Four techniques (DPPH, ABTS, CUPRAC, and FRAP) were used to measure antioxidant activity. Table 1 displays the ranges of quercetin (0/25 kGy), rutin (0/25 kGy), and vitamin C concentrations prepared for the study.

To make a solution of the radical (for the DPPH assay), 3.9 mg of DPPH was dissolved in 50 mL of methanol. For nearly two hours, the solution was shaken in the dark. To make an ABTS solution, 7.0 mM ABTS and 2.45 mM aqueous potassium persulfate (1:1 *v*/*v*) were mixed in water. For around 24 h, the solution was incubated in the dark. Once the absorbance reached 0.77, it was diluted with deionized water (measured at 734 nm). Neocuproine (7.5 × 10^−3^ M), copper (II) chloride (10.0 mM), and ammonium acetate buffer (pH 7.0) were all mixed together to create the CUPRAC solution. Next, 25 mL of acetate buffer (pH = 3.6), 2.4 mL of TPTZ solution, and 2.5 mL of a 20 mM aqueous FeCl_3_∙6H_2_O solution were mixed to make the FRAP test solution.

The working solution and sample solution were placed in a 96-well plate (6 replicates for each concentration). The plate was then covered with foil, shaken, and incubated at 37 °C (FRAP assay) or room temperature (DPPH/ABTS/CUPRAC assay). A Multiskan GO UV reader was used to determine color changes (Thermo-Scientific, Waltham, MA, USA). There were two measurements made, and I standard utilized was ascorbic acid. Table 2 displays the key parameters for each technique.

The sample’s level of radical scavenging for the DPPH and ABTS assay was determined using Equation (2):(2)$$the\;degree\;of\;radical\;scavenging\;(\%)\;=\;\frac{A_{0}-A_{i}}{A_{0}}\cdot 100\%$$
where $A_{0}$ is the absorbance of the control and $A_{i}$ is the absorbance of the sample.

We present the results of the DPPH and ABTS assay as a plot of %inhibition vs concentration, and the results of the CUPRAC and FRAP assay are presented as a plot of absorbance vs concentration. The IC_50_ or IC_0.5_ value was determined from linear or polynomial regression analysis. X (final sample concentration) for IC_50_ was calculated when Y in the regression equation was substituted with 50. For IC_0.5_, Y was substituted with 0.5.

## 5. Conclusions

An official sterilization technique of drugs that is accepted by the IAEA and listed in pharmacopeias is radiation sterilization. Analysis of the structure of rutin and quercetin after exposure to ionizing radiation at a dose of 25 kGy makes it possible to rule out their decomposition. In addition, we confirmed that ionizing radiation does not change the chemical structure of tested flavonoids and their antioxidant properties. The obtained results confirmed that radiation sterilization using an electron beam is a safe method of sterilizing rutin and quercetin in the solid state. The use of an electron beam as a method of sterilization of plant-derived compounds will make it possible to provide patients with safe and effective therapy using preparations with recognized antioxidant potential.

## Figures and Tables

**Figure 1 molecules-28-02713-f001:**
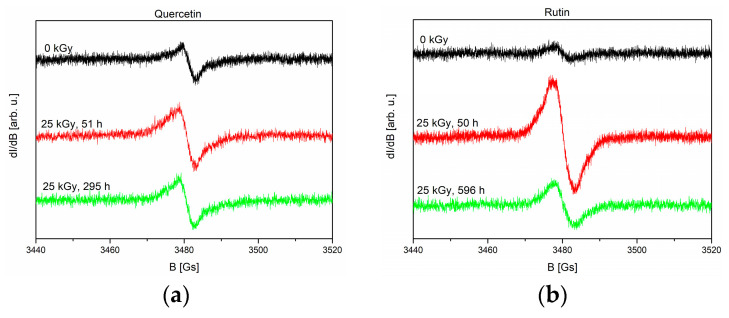
EPR spectra of non-irradiated (0 kGy) and irradiated (dose 25 kGy) samples recorded 51 and 595 h (quercetin) (**a**); 50 and 596 h (rutin) (**b**) after irradiation. The concentration of free radicals vs. time after irradiation (25 kGy) calculated from EPR spectra for quercetin (**c**) and rutin (**d**), respectively. The solid lines in (**c**,**d**) are the approximations of Equation (2) to the experimental points.

**Figure 2 molecules-28-02713-f002:**
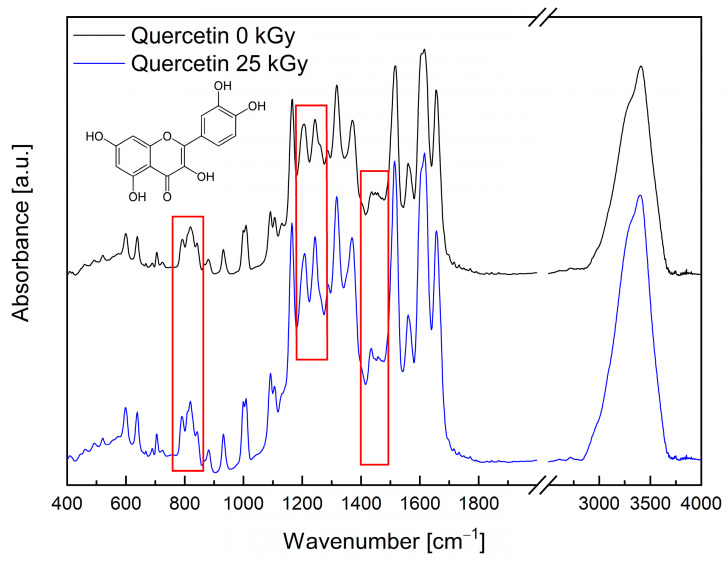
IR absorption spectra of non-irradiated (Quercetin 0 kGy—black) and irradiated (Quercetin 25 kGy—blue) quercetin at room temperature, range from 400 to 4000 cm^−1^.

**Figure 3 molecules-28-02713-f003:**
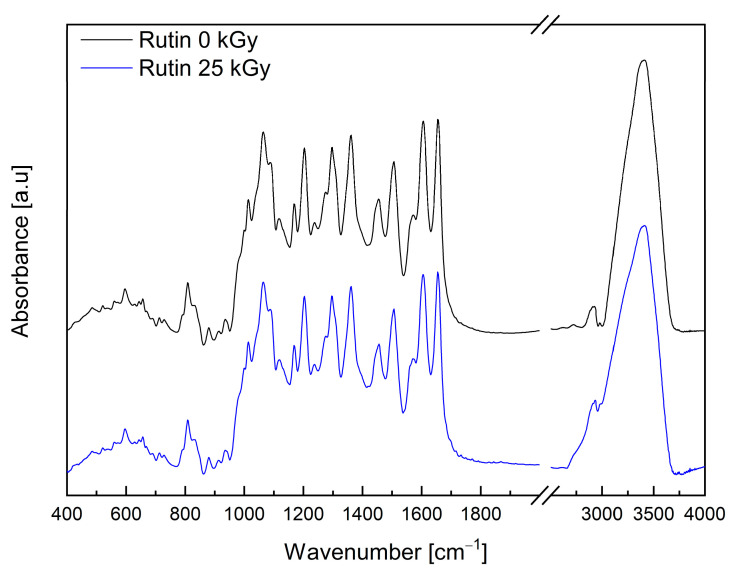
IR absorption spectra of non-irradiated (Rutin 0 kGy—black) and irradiated (Rutin 25 kGy—blue) rutin at room temperature, range from 400 to 4000 cm^−1^.

**Figure 4 molecules-28-02713-f004:**
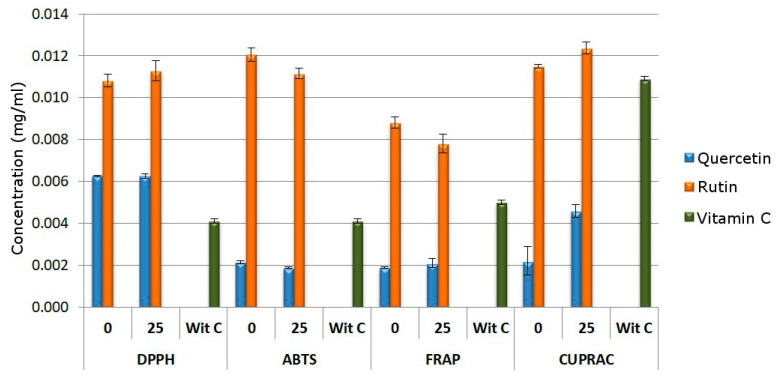
Summary of the results of antioxidant properties. IC_50_ value in DPPH and ABTS assay and IC_0.5_ value in FRAP and CUPRAC assay.

**Table 1 molecules-28-02713-t001:** Ranges of quercetin, rutin, and ascorbic acid concentrations used in studies of antioxidant properties.

Method	Quercetin [μg·mL^−1^]	Rutin [μg·mL^−1^]	Ascorbic Acid [μg·mL^−1^]
DPPH assay	12.5–100.0	25.0–150.0	10.0–100.0
ABTS assay	12.5–150.0	200.0–800.0	10.0–100.0
CUPRAC assay	25.0–100.0	12.5–100.0	8.0–125.0
FRAP assay	10.0–35.0	10.0–100.0	100.0–300.0

DPPH—2,2-difenylo-1-pikrylohydrazyl; ABTS—2,2′-azino-bis(3-ethylbenzothiazoline-6-sulfonic acid; CUPRAC—cupric reducing antioxidant capacity; FRAP—ferric reducing antioxidant power.

**Table 2 molecules-28-02713-t002:** The key parameters of DPPH, ABTS, CUPRAC, and FRAP effects.

Method	Sample Solution + Working Solution	Incubation	Measured
DPPH	25 µL + 175 µL	30 min. reaction, 5 min.: 600 rpm, 25°C	517 nm
ABTS	10 µL + 200 µL	10 min. reaction, 10 min.: 600 rpm, 25°C	734 nm
CUPRAC	50 µL + 150 µL	30 min. reaction, 5 min.: 600 rpm, 25°C	450 nm
FRAP	25 µL + 175 µL	30 min. reaction, 30 min.: 100 rpm, 37°C	593 nm

DPPH—2,2-difenylo-1-pikrylohydrazyl; ABTS—2,2′-azino-bis(3-ethylbenzothiazoline-6-sulfonic acid; CUPRAC—cupric reducing antioxidant capacity; FRAP—ferric reducing antioxidant power.

## Data Availability

The data are contained within the article and Appendix A.

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
