# Peer review of "Do Rutin and Quercetin Retain Their Structure and Radical Scavenging Activity after Exposure to Radiation?"

_molecules, 2023, doi:10.3390/molecules28062713_

Round 1
Reviewer 1 Report
row 55-56 please rewrite the sentence
it is important to express the aims of your work more clear e.g. we sbmited the rutin to a radiation source... in order to find out/verify....the supposition is.....
it is also important to formulate in the Conclusions chapter the importance of your findings and the future of your work
Reviewer 2 Report
This is a simple and straightforward study that aimed to assess if quercetin and rutin tolerate irradiation sterilization, as recommended by international guidelines. The introduction covers key literature and present enough background. The experimental design in scientifically sound. Conclusions are supported by data. There are, however, some minor issues that should be addressed:
1. I believe the title must be revised. In addition to English language, no biological activity at all was measured. Only in vitro chemical-based assays were used. Thus, the authors should not refer to biological activity in the title. A representative title would be: “Do Rutin and Quercetin retain their structure and radical scavenging activity after exposure to radiation?” or, in an affirmative form, “Rutin and Quercetin retain their structure and radical scavenging activity after exposure to radiation”.
2. The abstract can benefit from a concluding take home message statement.
3. Line 33: I would add that polyphenols might also chelate metals and reduce metals.
4. Line 39: The authors should distinguish two very different scenarios: (i) the exposure of a plant to stressful conditions leading to different production of bioactive molecules, and (ii) the irradiation of purified/isolated compounds changing their properties. These are completely different situations.
5. Line 49: Fagopyrum esculentum should in italics.
6. Lines 71–72: Include a citation/reference to support this statement.
7. The authors must provide EPR data for the compounds before any irradiation (i.e., 0 kGy), as they did for FTIR. This is essential for the understanding of the effects of radiation.
8. Line 243: Use a bullet point instead of a square to denote the unpaired electron in the superoxide molecule.
9. Line 294: Please provide more details about the irradiation procedure. Were Quercetin and Rutin irradiated in powder? In solution? Which concentration and solvent? Was any container used?
Reviewer 3 Report
Dear authors,
In the article proposed by Rosiak and colleagues, the biological stability of Rutin and Quercetin was examined after exposure to Radiation. The authors are aiming to determine the effects of ionizing radiation (electron beam radiation of 107 25 kGy dose) on the antioxidant properties and stability of both compounds.
With all respect to the efforts spent on this work, several comments noticed:
1- The introduction section has several mistakes regarding the examined compounds. Some compounds' description, like structure and mechanism of action , is not clear and need to be revised carefully.
2- The introduction section is too wordy and unhelpful. We do not need to read a litany of information in the introduction.
A- The references and their relativity to their work need to be revised carefully. For example, the author mentioned the role of the presence of C6-C3-C6 moieties and the presence of hydroxyl groups in the polyphenols, which reported reference number 7. I couldn’t find the relation.
B- The whole manuscript needs to be revised carefully, and several points need to be noticed:
· Polyphenols are a class of secondary metabolites, several subclasses that need to be known when the authors want to describe the SAR and MOA of the compounds of the mentioned flavonoids.
· It is not necessary to write in detail about what other literature's methodology and results are in the introduction; however, it is sufficient to mention briefly what is only related to the subject.
In conclusion, I couldn’t approve this manuscript with its current status, and this level of findings can be published as short communication after it is revised.
Good luck
Round 2
Reviewer 3 Report
Dear Author,
Thank you for extensive revision of your article. It seems more appropriate for publication.
Good luck